# Linked Color Imaging for Stomach

**DOI:** 10.3390/diagnostics13030467

**Published:** 2023-01-27

**Authors:** Eiji Umegaki, Hiraku Misawa, Osamu Handa, Hiroshi Matsumoto, Akiko Shiotani

**Affiliations:** Department of Gastroenterology & Hepatology, Kawasaki Medical School, 577 Matsushima, Kurashiki-shi, Okayama 701-0192, Japan

**Keywords:** endoscopic imaging, image-enhanced endoscopy, linked color imaging, gastritis, gastric intestinal metaplasia, early gastric cancer, the International Commission on Illumination (CIE) 1976 (L∗a∗b∗) color space, diffuse redness, lavender color sign

## Abstract

Image-enhanced endoscopy (IEE) plays an important role in the detection and further examination of gastritis and early gastric cancer (EGC). Linked color imaging (LCI) is also useful for detecting and evaluating gastritis, gastric intestinal metaplasia as a pre-cancerous lesion, and EGC. LCI provides a clear excellent endoscopic view of the atrophic border and the demarcation line under various conditions of gastritis. We could recognize gastritis as the lesions of the diffuse redness to purple color area with LCI. On the other hand, EGCs are recognized as the lesions of the orange-red, orange, or orange-white color area in the lesion of the purple color area, which is the surround atrophic mucosa with LCI. With further prospective randomized studies, we will be able to evaluate the diagnosis ability for EGC by IEE, and it will be necessary to evaluate the role of WLI/IEE and the additional effects of the diagnostic ability by adding IEE to WLI in future.

## 1. Introduction

The chromo-endoscopy method has long been used for improving the accuracy of endoscopic diagnosis. For instance, by dying with indigo-carmine, we can recognize the mucosal surface structure and the lesion boundary.

Since the early 2000s, image-enhanced endoscopy (IEE) has been developed and reported to improve the detection rate and the diagnostic accuracy of gastrointestinal lesions. IEE techniques are more useful than white-light imaging (WLI) for enhancing the images of surface patterns, vascular patterns, and color differences, as well as for detecting the demarcation line between the lesions and the surround mucosa [1,2,3,4,5,6,7,8,9,10,11,12,13].

Object-oriented classification for endoscopic imaging was described, and precise definitions of individual terms were provided from Japan in 2009 [12] (Figure 1).

We regularly utilize these modalities and try to improve the accuracy of diagnosis on a daily basis. It is highly important to recognize *Helicobactor pylori* (*H. pylori)*-associated gastritis and evaluate the spread of intestinal metaplasia, which is a high-risk pre-condition of gastric carcinoma. Furthermore, diagnosing gastric carcinoma at its early stage is beneficial to improving quality of life for individuals.

In this review, we will focus on the diagnosis of gastritis, gastric intestinal metaplasia (GIM), and early gastric cancer (EGC) with LCI, and we will assess the value and future prospects of LCI.

## 2. Basic of LCI Examination

Recently, multilight imaging technology using narrow-band imaging (NBI, Olympus Co., Tokyo, Japan), flexible spectral color enhancement (FICE, Fujifilm Co., Tokyo, Japan), blue laser imaging (BLI, Fujifilm Co., Tokyo, Japan), linked color imaging (LCI, Fujifilm Co., Tokyo, Japan) and i-SCAN optical enhancement (i-SCAN, PENTAX Co., Tokyo, Japan) has been developed to resolve the limitations of diagnosis by the conventional endoscopy [1,2,3,4,5,6,7,8,9,10].

The characteristics of LCI mode and WLI mode are shown in Table 1 [4]. The emission intensity of the short wavelength release in LCI is much higher than that of WLI.

LCI detects the mucosal surface patterns clearly including the microstructure because of the high intensity at short wavelengths including 410 nm and 450 nm. With a wavelength of 410 nm, both the microstructure and microvasculature in the superficial layer of the mucosa are emphasized. Digital signal/image processing in LCI systems emphasizes slight color differences and provides a better color contrast within the red color range. Consequently, the regions that were originally red appear redder, and regions that were originally white appear whiter, but with natural tones. LCI shows a mucosal color similar to WLI and also produces different color patterns of the mucosa from WLI due to emission intensity at certain wavelengths. LCI is brighter than WLI and useful for the screening of gastrointestinal lesions even as a distant view from the large lumen such as the stomach. We were able to obtain several color images and detailed information; micro-surface patterns and micro-vascular patterns were represented using the LCI mode, although this was not possible with the WLI mode.

LCI was able to emphasize tiny differences in color, which is due to differences in reflectance from the superficial layer of the mucosa at short wavelengths. These differences may occur from various architectural differences, histological differences, mucosal blood flow, mucous content, and so on in the gastrointestinal mucosa [13].

By irradiating the narrowband light, high contrast images of the surface mucosa can be obtained. The slight difference in color of close to a mucosal color is emphasized by the difference between illumination light and of signal processing. As a result, the originally red regions appear redder, and originally white regions appear whiter, but with natural tones. The method could be useful for visualizing reddish/brownish areas of the mucosa and detecting inflammation and minute changes in the mucosa. LCI would be useful for detection with micro-surface patterns and micro-vascular patterns in the mucosa. Slight color differences are visualized with natural tones.

In Japan, both laser light source (Laser) endoscopy and light-emitting diode light source (LED) endoscopy are available. When using either laser endoscopy or LED endoscopy, LCI (Laser-LCI/LED-LCI) was found to be more effective than WLI (laser-WLI/LED-WLI) for detecting EGCs and evaluating *H. pylori*-associated gastritis [11].

## 3. LCI for Gastritis (*Helicobacter pylori* Infection)

### 3.1. Endoscopic Images of LCI for Gastritis

A number of studies on LCI have been published. LCI was found to have better detection rates, sensitivity, and specificity for GI lesions compared with WLI without significant difference in the time taken for endoscopic examination with LCI [14].

It is highly important to recognize the chronic active gastritis induced by *H. pylori* infection and the spread of the atrophic mucosa, which is the high-risk pre-condition of gastric carcinoma.

Dohi, O. and Iwamura, M., et al. reported that *H. pylori* infection can be identified by enhancing endoscopic images of the diffuse redness using LCI; furthermore, LCI is useful for diagnosing current gastric *H. pylori* infection [15,16]. Chen, T.H., et al., also termed it the diffuse redness to purple color under LCI [17].

### 3.2. Usefulness of LCI for Gastritis

Endoscopic color features with LCI can help to diagnose *H. pylori*-associated gastritis and evaluate the spread of the mucosal lesions [18,19,20,21,22].

Mizukami, K., et al., reported that LCI provides a clear endoscopic view of the border between an atrophic mucosa and a non-atrophic mucosa under various conditions of gastritis, regardless of *H. pylori* infection status [23]. Lee, S.P. et al., reported on the better diagnostic accuracy of LCI for *H. pylori* infection status [20]. On the other hand, Ono, S. et al. reported that LCI is useful for the endoscopic diagnosis of *H. pylori*-active or inactive gastritis, and it is advantageous for patients with past infections of inactive gastritis [24].

Jiang, Z.X. et al. assessed the identification of *H. pylori*-associated gastritis with the LCI score, which is composed of the endoscopic images showing redness of fundic glands, granular erosion, purple mucus, and mucus lake turbidity by the LCI mode of endoscopy. This LCI score showed high sensitivity and specificity for the differential diagnosis of *H. pylori*-associated gastritis, and it is an effective method for identifying *H. pylori* infection in gastric mucosa [25]. In the case with magnifying endoscopy, the diagnostic rate with the LCI mode was higher for *H. pylori* infection than with any other IEE [26].

In a prospective randomized tandem gastroscopy pilot study of LCI versus WLI imaging for the detection of upper gastrointestinal lesions, LCI had better detection rates, sensitivity, and specificity for detection of upper GI lesions, atrophic gastritis, and intestinal metaplasia compared to WLI [14].

In Figure 2a, we show typical images of the FP-border (atrophic/non-atrophic border) in a patient with chronic active gastritis. With LCI, we can recognize its border easily.

Furthermore, the area of active gastritis is identified by enhancing endoscopic images of the diffuse redness to purple with LCI (Figure 2b).

### 3.3. Visibility of Each Endoscopic Finding of the Kyoto Classification by LCI

Conventional endoscopy is not suitable for diagnosing the spread of atrophic mucosa and intestinal metaplasia; therefore, it remains mandatory that a biopsy is performed to facilitate a histopathological assessment of the gastric mucosa with the updated Sydney classification. However, IEE improved the accuracy and reproducibility of endoscopic findings of pre-cancerous gastric lesions.

In the Kyoto global consensus report on Helicobacter pylori induced gastritis, appropriate diagnostic assessment of gastritis was proposed [27]. The visibility of endoscopic features, including diffuse redness, spotty redness, map-like redness, patchy redness, red streaks, intestinal metaplasia, and an atrophic border, were assessed with LCI and WLI [28]. Compared with WLI, all endoscopists reported improved visibility with LCI: 55.8% for diffuse redness; LCI: 38.5% for spotty redness; LCI: 57.7% for map-like redness; LCI: 40.4% for patchy redness; LCI: 53.8% for red streaks; LCI: 42.3% and BLI-bright: 80.8% for intestinal metaplasia; and LCI: 46.2% for an atrophic border. The visibility of each endoscopic finding of the Kyoto Classification was improved with LCI, while that of intestinal metaplasia was improved with BLI-bright compared with WLI.

The review of Dohi, O. et al. focuses on understanding clinical applications and corresponding evidence. The accuracy of the diagnosis for gastritis based on the Kyoto classification was found to be improved with the currently available advanced technologies of various IEEs, including LCI, NBI, and BLI [29].

### 3.4. Evaluation of LCI for Gastritis with CIE 1976 (L* a* b*) Color Space

In order to evaluate a color difference objectively, the International Commission on Illumination (CIE) 1976 (L*a*b*) color space is used, and the color difference (ΔE) is calculated. The CIE 1976 (L*a*b*) color space is a three-dimensional model composed of a black to white axis (L*), a green to red axis (a*), and a blue to yellow axis (b*). L* defines brightness, a* defines the green-red component, and b* defines the blue-yellow component. This color space is designed to approximate human perceptions, and the Euclidean distance between two points is proportional to the difference in perception of the corresponding colors [30] (Figure 3).

Mizukami, K. et al. reported the color values of atrophic and non-atrophic mucosa with The CIE 1976 (L*a*b*) color space. Color differences at the atrophic boundary, defined as Euclidean distances of color values between the atrophic and non-atrophic mucosa, were compared with WLI / LCI for the overall cohort and separately for patients with *H. pylori* infection. The color difference became significantly higher with LCI than with WLI in all cases, and LCI made the endoscopic view of the atrophic mucosal boundary with surrounding mucosa clearer under various conditions of gastritis, regardless of *H. pylori* infection [23].

Sakae, H. et al. evaluated the color changes of the gastric mucosa with the CIE 1976 (L*a*b*) color space after *H. pylori* eradication therapy. At the gastric body, the a* value, which reflected a green to red axis, decreased significantly after *H. pylori* eradication therapy with LCI and WLI. The a* values were generally associated with histopathological neutrophil infiltrations, and quantitative evaluation revealed that LCI emphasizes the change in color of the gastric mucosa due to the reduction in diffuse redness. On the other hand, there were no significant differences with either mode at the antrum [31].

For transnasal thin endoscopy, the color difference at the fundic/pyloric mucosal border, in patients with current and past *H. pylori* infection, was greater with LCI than with WLI. LCI effectively detects the endoscopic atrophic border [32].

### 3.5. AI with LCI for Gastritis

Deep learning is a type of artificial intelligence (AI) that imitates the neural network in the brain. Nakashima, H. et al. generated an AI method to diagnose *H. pylori* infection with BLI-bright and LCI. The developed AI method had an excellent ability to diagnose *H. pylori* infection with BLI-bright and LCI. AI technology with IEE is likely to become a useful image tool for diagnosing gastritis, GIM, and EGC [33]. Following this, a computer-aided diagnosis (CAD) system based on LCI combined with deep learning was developed. The diagnostic accuracy of the LCI-CAD system was 84.2% for *H. pylori* uninfected status, 82.5% for *H. pylori* infected status, and 79.2% for post *H. pylori* eradication status. These results revealed better accuracy of diagnosis based on the LCI-CAD system relative to the WLI-CAD for *H. pylori* uninfected, *H. pylori* currently infected, and post *H. pylori* eradication cases. Furthermore, there were no differences in diagnostic accuracy between the LCI-CAD system and experienced endoscopists with the validation data set of LCI. This study shows the feasibility of a gastric cancer screening program to determine cancer risk in individual subjects based on LCI-CAD [34].

Yasuda, T. et al. constructed an interpretable automatic diagnostic system using a support vector machine for *H. pylori* infection and compared the diagnostic ability of its AI system with that of endoscopists. The presence of *H. pylori* infection determined with LCI was learned through machine learning. The accuracy of the AI system was higher than that of an un-experienced endoscopist, but there was no significant difference between the diagnosis of experienced endoscopist and the AI system. As a result, this AI system could diagnose *H. pylori* infection with significant accuracy and provide more useful diagnostic information by learning images and considering the diagnostic algorithm for pre-/post-*H. pylori* eradication [35].

## 4. LCI for Gastric Intestinal Metaplasia

### 4.1. Endoscopic Images of LCI for Gastric Intestinal Metaplasia

Recently, there have been various reports that IEE may improve the detection rate of gastric intestinal metaplasia (GIM) [36,37,38]. GIM is generally thought to be the mucosal background for the development of gastric adenocarcinoma. Therefore, it is important to diagnose GIM with endoscopy.

Ono, S. et al. reported that GIM is recognized as a lavender area with noninvasive LCI examination, referred to as the “Lavender color sign” (LCS). The diagnostic accuracy of target biopsies for GIM was 23.7% with WLI and 84.2% with LCI. The LCI mode could provide a new diagnostic tool for detecting GIM during routine endoscopy [39,40,41].

Min, M. et al. reported that the color pattern in areas of GIM was purple mixed with white on the epithelium with signs of mist that were detected during non-magnifying LCI observations. They termed this endoscopic finding “Purple in Mist” (PIM). For the diagnosis of GIM, compared to histopathological assessment, the LCI finding had an accuracy of 91.1% (95%CI: 86.5–95.7%), a sensitivity of 89.8% (95%CI: 81.3–98.3%), a specificity of 91.8% (95%CI: 86.3–97.2%), a positive predictive value of 84.6% (95%CI: 74.8–94.4%), and a negative predictive value of 94.7% (95%CI: 90.1–99.2%) [42].

With the LCI mode, Chen, H. et al. tested whether the specific color feature of “patchy lavender color” (PLC) pathologically indicated GIM or not. The diagnostic accuracy rate for GIM by LCI was 79.44%, which is higher than that of WLI (40.19%) (*p*  <  0.001). Moreover, LCI with targeted biopsies showed a significantly increased ability to detect GIM (*p*  <  0.001). PLC observed in the gastric mucosa with the LCI mode could guide endoscopic biopsies and increase the detection rate of GIM [43].

“Lavender color sign” (LCS), “Purple in Mist” (PIM), and “patchy lavender color” (PLC) are regarded here as similar terms indicating GIM with LCI mode examination.

We present a case of atrophic gastritis after *H. pylori* eradication therapy (Figure 4). With WLI, we recognized the map-like redness at the lesser curvature of the gastric body, and with LCI, we recognize its lesion as the purple color area with the clear boundary against surrounding mucosa.

### 4.2. Gastric Intestinal Metaplasia and Gastric Carcinoma

GIM is found in the atrophic mucosa of chronic gastritis induced by *H. pylori* infection and is generally considered to be the pre-cancerous mucosal lesions in the development of gastric adenocarcinoma. These GIM are recognized as a purple color area with LCI examination. Fukuda, H. et al. enrolled fifty-two patients with early gastric carcinoma and calculated color differences between malignant lesions and surrounding mucosa prospectively; they compared the obtained results with histopathological findings in resected specimens. Carcinoma and surrounding mucosa in 74% of lesions had similar colors with the WLI mode, whereas purple mucosa surrounded part of the whole carcinoma appearing as orange-red, orange, or orange-white in the LCI mode. The surrounding purple area in the LCI mode corresponded histopathologically to GIM. Endoscopic images in the LCI mode have a higher color contrast between EGC and the surrounding mucosa compared to in the WLI mode [44].

Majima, A. et al. enrolled individuals with newly detected gastric cancer (GC) after *H. pylori* eradication therapy (CA group, *n* = 109) and individuals without GC (NC group, *n* = 85), and evaluated the endoscopic findings of the background mucosa with WLI and LCI, based on the Kyoto classification of gastritis. Map-like redness and severe atrophy were significantly more frequent in the CA group than in the NC group. Map-like redness, which reflects GIM after successful *H. pylori* eradication therapy, was identified more frequently with LCI than WLI, and the absence of regular arrangement of collecting venules (RAC) was associated with GC detected after *H. pylori* eradication therapy [45].

In a systemic review and meta-analysis conducted by Shu, X. et al., a total of six original studies involving 700 participants were included, and the sensitivity, specificity, positive likelihood ratio, and negative likelihood ratio of LCI for diagnosing GIM were 0.87 (0.83–0.91), 0.86 (0.82–0.89), 5.72 (3.63–8.99), and 0.17 (0.08–0.36), respectively. LCI could be used for the accurate diagnosis of GIM, which is a high-risk condition, a pre-cancerous lesion for intestinal-type gastric carcinoma. Considering the weaknesses of the available studies in terms of design, further studies with a rigorous design are needed for further validation of the findings in this article [46].

## 5. LCI for Gastric Carcinoma

### 5.1. Usefulness of LCI for Gastric Carcinoma

Various kinds of IEE have been reported to improve the detection rate and the diagnostic accuracy of gastrointestinal lesions [47,48,49,50,51,52,53,54,55]. IEE techniques are useful for enhancing images of mucosal vasculature patterns and for improving the visibility of surface mucosal patterns and color differences [1,2,3,4,5,6,7,8].

In a case report, Ono, S. reported the advantage of LCI during routine endoscopy. GIM, which is a high-risk condition for gastric carcinoma, was easily detected in the LCI mode as a lavender color. Furthermore, use of the LCI mode enhanced a gastric carcinoma in GIM. LCI would be a useful tool for detecting gastric carcinoma in high-risk patients [56]. Fukuda, H. et al. also reported that flat EGCs became clearly visible in the LCI mode [47].

In a review titled “Current *Helicobacter pylori* Diagnostics”, the ability to diagnose inflammation and atrophy of the mucosal surfaces, in order to facilitate diagnosis of EGC, was found to have improved significantly due to the development of new endoscopic technologies in the diagnosis of *H. pylori* infection and EGC, such as BLI, LCI, and magnifying endoscopy [57].

Yoshifuku, Y. et al. reported that LCI improved the visibility of EGC, regardless of the ability of the endoscopist or *H. pylori* eradication therapy in patients, particularly for EGC with a reddish or whitish color [58]. LCI significantly improves visibility of EGC regardless of differences in lesion morphology, location, depth of invasion, histology, and *H. pylori* infection status compared to conventional WLI [59,60].

Fockens, K. et al. evaluated the additional effect of LCI next to WLI for the identification of EGC. The addition of LCI next to WLI improves the visualization of EGC. Experts reach a higher consensus on discrimination between neoplasia and inflammation when using LCI. Non-experts improve their targeted biopsy placement with the use of LCI. LCI therefore appears to be a useful tool for the identification of EGC [53].

In addition, there are a number of other reports that described the usefulness of LCI for detecting EGCs. After *H. pylori* eradication therapy, the border between EGC and the surrounding mucosa becomes indistinct, and the diagnosis of EGC becomes difficult. Nevertheless, LCI significantly improved the visibility of EGCs after *H. pylori* eradication therapy compared with WLI. Furthermore, LCI significantly reduced miss rates of these lesions compared with WLI [61]. The detection rate of gastric neoplastic lesions was higher in the LCI + WLI group than in the WLI group; therefore, LCI might be an effective method for screening early gastric carcinoma [62].

LCI is superior to WLI for detecting EGCs with atrophic gastritis [63].

With magnified endoscopy, Kitagawa, Y. et al. reported that magnifying linked color imaging with indigo carmine dye (M-Chromo-LCI) facilitates sterically enhanced and color image-magnified observations of the superficial gastric mucosa, and it is expected to become a useful modality for the accurate diagnosis of gastric lesions [64].

### 5.2. LCI for Gastric Carcinoma with Ultrathin Endoscopy

Ultrathin endoscopy causes a minimal gag reflex and has minimal effects on cardiopulmonary function. LCI is useful for the detection of gastritis, GIM, and malignancies in the digestive tract. Khurelbaatar, T. et al. reported that the LCI mode with a low-resolution ultrathin endoscope is superior to the WLI mode with a high-resolution standard endoscope for gastric carcinoma screening. The superior high color contrast between EGC and the surrounding mucosa may be more important than high resolution images [65].

LCI is a new image-enhancing technique that facilitates the differentiation of slight differences in mucosal color tone. Haruma, K. et al. performed an exploratory analysis to evaluate the diagnostic ability of the LCI mode in ultraslim endoscopy using data from patients examined in the LCI-Further Improving Neoplasm Detection in upper gastrointestinal (LCI-FIND) trial [10]. Ultraslim endoscopes were used in 223 patients, and standard endoscopes were used in 1279 patients; the percentage of patients diagnosed with a neoplastic lesion tended to be higher in the LCI mode than with the WLI mode among patients who underwent ultraslim endoscopy and among those who underwent standard endoscopy. LCI is useful in identifying neoplastic lesions, even when used in ultraslim endoscopy [66].

We show a case of early gastric carcinoma (Type 0–IIc) with atrophic gastritis (Figure 5). With WLI, we recognized the minute shallow depressed lesion at the anterior wall of the gastric antrum; with LCI, its lesion becomes clear as the orange-red color area with a clear boundary against surround mucosa.

### 5.3. Evaluation of LCI for Gastric Carcinoma with CIE 1976 (L* a* b*) Color Space 

Kanzaki, H., et al., evaluated the color differences between the lesion and the surrounding mucosa (ΔE) using the CIE L*a*b* color space. The average ΔE values with LCI, BLI-bright, and WLI were 11.02, 5.04, and 5.99, respectively. The ΔE was significantly higher with LCI than with WLI (*p* < 0.001). LCI led to a larger ΔE than WLI, and it may provide easy recognition and early detection of gastric carcinoma, even for less-experienced endoscopists [67].

Dohi, O. et al. also reported that the median ΔE values with WLI for gastric neoplasms missed under WLI and later detected under LCI were significantly lower than those for gastric neoplasms detected under WLI (8.2 vs. 9.6, respectively). Furthermore, low levels of ΔE with WLI (odds ratio [OR], 7.215) and high levels of ΔE with LCI (OR, 22.202) were significantly associated with gastric neoplasms missed under WLI and later detected under LCI [68].

Fukuda, H. et al. reported that LCI images have a higher color contrast between EGCs and surrounding mucosa compared to WLI images. A characteristic purple color around gastric cancers with LCI represents intestinal metaplasia, and orange-red, orange, or orange-white color with LCI represents gastric cancers [44]. In cancer lesions, the density of the surface blood vessel was significantly higher in comparison with surrounding non-cancer lesions. LCI is more effective for the recognition of EGC compared to WLI as a result of the improved visualization of changes in redness [69].

Kanzaki, H. et al. conducted a retrospective study of LCI of differentiated EGCs and suspicious mucosal areas to compare the color differences between each lesion. The mean color values of EGC by the CIE L*a*b* color space were as follows: L* (lightness), 61.7; a* (green to red), 41.2; and b* (blue to yellow), 27.1. Those of suspicious mucosal areas were as follows; L*, 56.1; a*, 44.2; and b*, 21.3. EGC had significantly higher L*, b* values and lower a* values in comparison to suspicious mucosal areas. EGC had higher b* values in comparison to suspicious mucosal areas and was not only reddish but also mixed with yellow, with an orange-like color [70].

In the near future, we may diagnose EGCs as a result of the color analysis in the CIE L*a*b* color space.

### 5.4. LCI for Stomach and Other Alimentary Tract

In a controlled, multicenter trial with randomization using minimization, the performance of LCI with WLI in detecting neoplastic lesions in the upper gastrointestinal tract was compared. In total, 752 patients were assigned to the WLI group, and 750 were assigned to the LCI group. The ratio of patients with one or more neoplastic lesions diagnosed in the first examination was higher with LCI than with WLI (60 of 750 patients or 8.0% vs. 36 of 752 patients or 4.8%; risk ratio 1.67). The proportion with overlooked neoplasms was lower in the LCI group than in the WLI group (five of 750 patients or 0.67% vs. 26 of 752 patients or 3.5%; risk ratio 0.19). LCI is more effective than WLI for detecting neoplastic lesions in the pharynx, esophagus, and stomach [10,68].

In routine upper/lower gastrointestinal endoscopy, Shinozaki, S. et al. reported that LCI examination improves the detection rate of colonic adenoma and decreases the overlook rate of the colonic polyp. LCI is useful for detecting superficial lesions throughout the gastrointestinal tract by enhancing the color contrast and the distribution of minute vessels between the lesion and the surrounding mucosa. LCI should be used in the routine upper and lower gastrointestinal endoscopy [71].

## 6. Future Prospects

IEE plays an important role in the detection and diagnosis of gastritis, GIM, and EGC. LCI is also useful for detecting and evaluating EGC. Yamaguchi, D. et al. used three IEE technologies, namely, NBI, BLI, and i-scan optical enhancement, and the detection rates for EGC between IEE and WLI were evaluated. The EGC detection rate in the IEE plus WLI groups was 77.3%. Although the EGC detection rate in the IEE group was higher than that in the WLI group (80.0% vs. 70.3%), there was no significant difference between these two modalities. An important detection factor using IEE was tumor circumference, where the rate of detection in the anterior wall and lesser curvature was significantly higher than that in the posterior wall and greater curvature. The detection rate of EGC without magnification was similar between the IEE group and the WLI group. Important factors for detecting EGC differed between IEE and WLI; therefore, the IEE mode and the WLI mode have different characteristics regarding EGC detection [72]. We also propose performing both WLI examination and IEE examination for the detection and evaluation of EGCs at this stage.

In a systematic review and meta-analysis titled “Diagnostic value of LCI based on endoscopy for GIM” by Shu, X. et al., LCI was found to be useful for the accurate diagnosis of GIM; however, considering the weaknesses of the available studies in terms of design, further studies with a rigorous design are needed for further validate the findings of this meta-analysis [46].

Various clinical studies using LCI are being conducted, and new evidence is being collected. By performing further prospective randomized studies, we will be able to evaluate the diagnostic ability for EGC by IEE; it will be necessary to evaluate the role of WLI/IEE and the additional effects of their diagnostic ability by incorporating IEE and WLI.

## Figures and Tables

**Figure 1 diagnostics-13-00467-f001:**
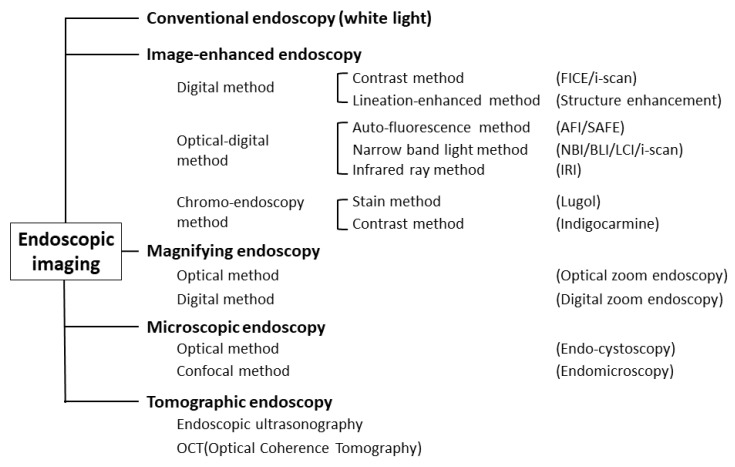
Classification of endoscopic techniques (adapted from Ref. [12]).

**Figure 2 diagnostics-13-00467-f002:**
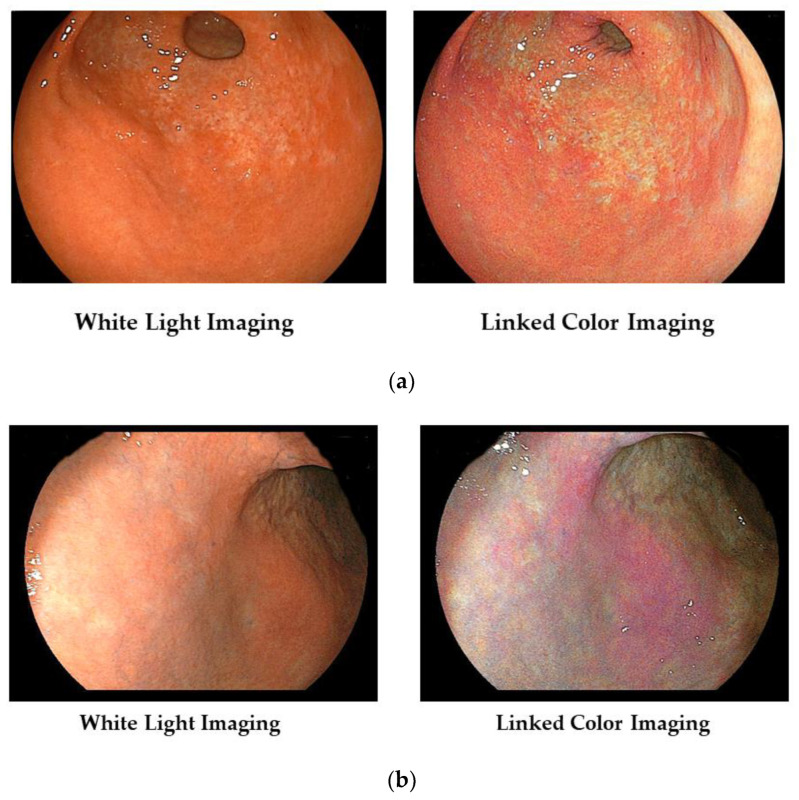
(**a**) Typical images of the FP-border (atrophic border) in a patient with chronic active gastritis. (**b**) Typical images of the *H. pylori*-associated active gastritis with WLI and LCI.

**Figure 3 diagnostics-13-00467-f003:**
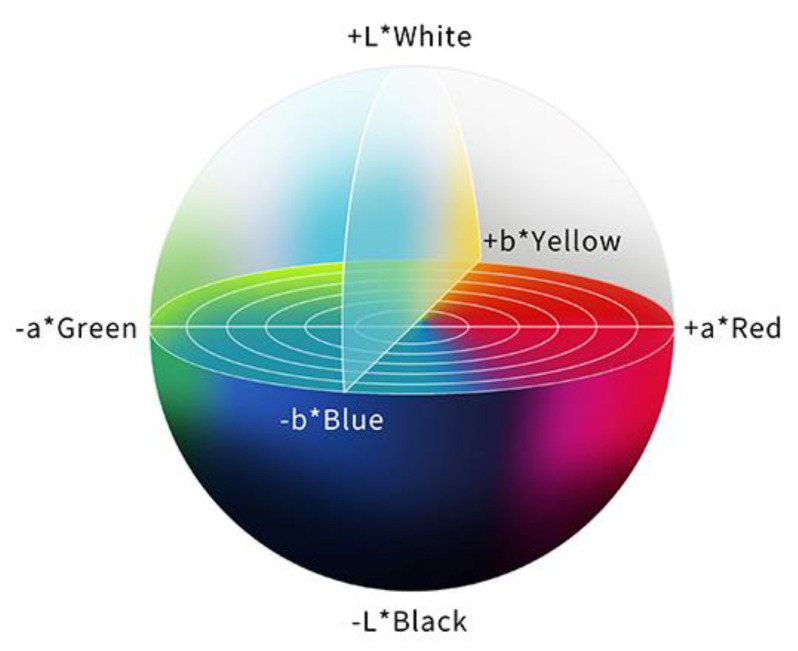
The color space according to the International Commission on Illumination 1976 definition (L*a*b*). This color space is designed to approximate human perceptions. The Euclidean distance between two points is proportional to the difference in perception of the two corresponding colors.

**Figure 4 diagnostics-13-00467-f004:**
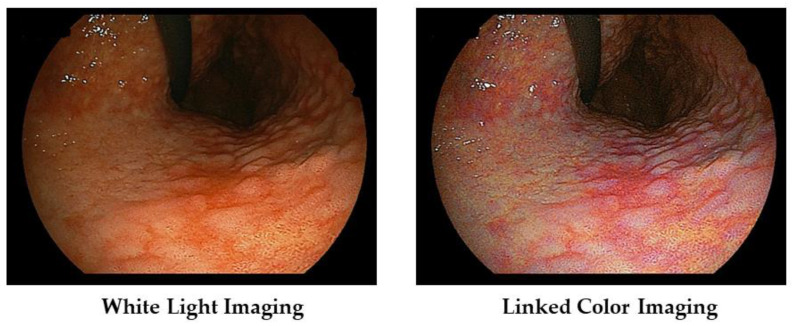
Typical images of atrophic gastritis (so called map-like redness) in a patient with post *H. pylori* eradication therapy.

**Figure 5 diagnostics-13-00467-f005:**
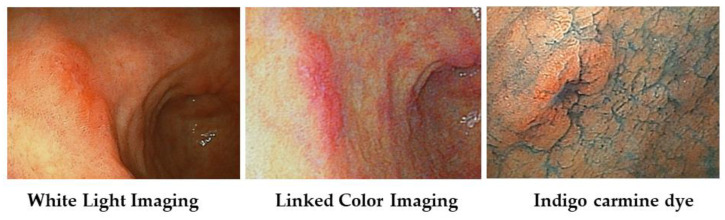
Case of early gastric carcinoma (Type 0–IIc) witn atrophic gastritis.

**Table 1 diagnostics-13-00467-t001:** Comparison of linked color imaging and white light imaging in the stomach (adapted from Ref. [4]).

	Linked Color Imaging	White Light Imaging
color information	Blue Green Red	Blue Green Red
Intensity at 410 nm (Violet blue)	High	Weak
Brightness	Brightest	Bright
Visibility of structures and vasculature	Clearly observed	Not clearly
Normal mucosa	Light orange	Coral
Intestinal metaplasia	Purple	White, Red
Cancer	Orange-red and others	Red, Discolored

## Data Availability

Not applicable.

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
