# Peer review of "Linked Color Imaging for Stomach"

_diagnostics, 2023, doi:10.3390/diagnostics13030467_

Round 1

Reviewer 1 Report

This article describes the importance of imaging techniques in the evaluation of gastric lesions I think that a more in-depth study of the specialized literature is necessary. I also identified numerous grammatical and text formatting mistakes. The indications regarding the formatting of references are not followed.  

Author Response

Thank you very much for reviewing my article. I corrected an article as follows.

1) Various clinical studies using LCI are being conducted, and new evidence is being collected. By performing further prospective randomized studies, we will be able to evaluate the diagnostic ability for EGC by IEE.

So I revised the text in ”Future Prospects”. (red in text)

2) For numerous grammatical and text formatting mistakes, I checked the text by the editing services:Language Editing Services with MDPI.

3) I followed the indications regarding the formatting of references and changed them.

Reviewer 2 Report

This the comprehensive review of identifying early gastric cancer technique, linked color imaging. I add some point to improve the manuscript:

1. Keywords should be limited in number.

2. It is better to expand the introduction more and discuss the technique and gastric cancer and the importance of detecting in early stage and its association with H.pylori in the first place. It is not necessary to mention Japan and...

3. It is better to mention tables or figures right after its reference in the manuscript, not at the end of the paper.

4. it seems that the tables or figures and their legends are copy paste. Make sure that they are your own unique table and figures.

5. The more images from patients, the better your review article becomes. Therefore, try to add more images from patients.

Author Response

Thank you very much for reviewing my article and your kindly suggestions.

I corrected an article as follows;

1.Three to ten pertinent keywords are accepted, so I choiced 8 keywords.

2.In introduction, I added that it was very importance to diagnose gastritis exactly and gastric carcinoma at the early stage by endoscopy. (red in text)

3.I changed the placement of Figures/Tables.

4.Table 1 is adapted from the reference, and I modified it.

Figure 1 is a copy paste, so I made a newly one.

The color space (L*a*b*) is defined in 1976, so I could not change Figure 3

5.I added the more images from patients. (Figure 2, 4)

Reviewer 3 Report

Suggestions:

1)     Line 9: Removing detailed with ‘in depth examination’

2)     Line 10: Needs full form because LCI  because I myself got confused and you're using this short form first time in the main content

3)     Line 18: should be diagnostic ability and not diagnosis ability

4)     Line 24: should use ‘for long time for diagnostic purposes’ and not ‘long time for diagnostic purposes’

5)     Line 37: should be ‘linked color imaging’ and not ‘linker color imaging’.

6)     Line 44: should be ‘to resolve’ and not ‘and resolve’.

7)     Line 48: should be ‘LCI detects’ and not ‘LCI detect’

8)     Line 51: what do you mean by ‘LCI he has another read information.’?

9)     Line 161: the ‘et al’ in front of the reference 39 does not make sense.

10)  Line 172: It should be ‘diagnostic ability’ and not ‘diagnosis ability’

11)  Line 184: it should be ‘it is very important’ not ‘it is a very important’

12)  I suggest more images of the endoscopies for the physicians to understand the difference between the visual activity of WLI and LCI.

13)  Line 264: It should be ‘surrounding’ and not ‘surroud’

14)  Line 309: It should be ‘surrounding’ and not ‘surroud’

15)  Line 334: it should be ‘detecting’ and not ‘detection’

16)  Line 356: It should be ‘needed’ and not ‘need’

17)  Line 358: It should be ‘diagnostic ability’ and not ‘diagnosis ability’

18)  Line 360: It should be ‘diagnostic ability’ and not ‘diagnosis ability’

Author Response

Thank you very much for your kindly suggestions.  I corrected them as you were pointed out.

About 8), I added explanation in detail. (red in text)

Digital signal/image processing in LCI systems emphasizes slight color differences and provides a better color contrast within the red color range. Consequently, the regions that were originally red appear redder, and regions that were originally white appear whiter, but with natural tones.

About 12), I added the more images from patients. (Figure 2,4)

Reviewer 4 Report

The authors of this review assessed the value and future prospects of linker color imaging (LCI) in diagnosing gastritis, gastric intestinal metaplasia, and early gastric cancer.

This paper is very well-written and informative. The references of this work are updated. 

Author Response

Thank you very much for reviewing my article.

Round 2

Reviewer 1 Report

The quality of the manuscript has been improved and its revised version warrants publication.